# Compressed Life Review: Extreme Manifestation of Autobiographical Memory in Eye-Tracker

**DOI:** 10.3390/bs10030060

**Published:** 2020-02-26

**Authors:** Veronika V. Nourkova

**Affiliations:** Department of Psychology, Lomonosov Moscow State University, Moscow 125009, Russia; Nourkova@mail.ru; Tel.: +7-903-573-4533

**Keywords:** compressed life review, panoramic memories, total recall, life-review experience, autobiographical memory, self-defining memories, eye-tracking, parallel awareness, working memory, long-term WM

## Abstract

The compressed life review (CLR) is a mnemonic illusion of having “your entire life flashing before your eyes”. This research was guided by concerns over the retrospective methodology used in CLR studies. To depart from this methodology, I considered the long-term working memory (WM), “concentric”, and “activation-based” models of memory. A novel theoretically rooted laboratory-based experimental technique aimed to elicit the CLR-like experience with no risk to healthy participants was developed. It consists of listening to superimposed audio recordings of previously trained verbal cues to an individually composed set of self-defining memories (SDMs). The technique evoked a self-reported CLR-like experience in 10 out of 20 participants. A significant similarity in eye movement patterns between a single SDM condition and a choir of SDM conditions in self-reported CLR experiencers was confirmed. In both conditions, stimuli caused relative visual immobilization, in contrast to listening to a single neutral phrase, and a choir of neutral phrases that led to active visual exploration. The data suggest that CLR-like phenomenology may be successfully induced by triggering short-term access to the verbally cued SDMs and may be associated with specific patterns of visual activity that are not reportedly involved with deliberate autobiographical retrieval.

## 1. Introduction

The compressed life review (CLR), also known as panoramic memories, total recall, replay of past experiences, or the life-review experience, is an intriguing mental phenomenon implying the extreme, yet instantaneous, manifestation of autobiographical memory (AM). Although a commonly accepted model of the CLR is lacking, further systematic investigations might be beneficial in order to understand AM mechanisms and functions. As survivors are the main source of information on CLRs, this hints that the phenomenon might be of a high adaptive value in helping to perform rescue actions [1,2]. Therefore, CLR studies have real-world implications, as they are in search of new avenues to assist people in stressful life circumstances.

Dozens of self-reported descriptions exist of an experience that may be summarized by the idiom ‘‘my whole life flashed before my eyes”. The earliest reports of this kind were published in the 19th century. For instance, one of the earliest examples was from Albert Heim, a Swiss geologist, who recollected his experience during a fall of 70 ft from a cliff face in the Alps. He wrote: “…I saw my whole past life take place in many images, as though on a stage at some distance from me. I saw myself as the chief character in the performance…” [3].

Some components of CLRs are associated with self-reports after life-threatening situations in which death seemed very probable: serious traffic accidents, mountaineering accidents, and shipwrecks [4,5]. Sometimes, a CLR coincides with a so-called near-death experience (NDE), i.e., an umbrella term for atypical states of mind occurring during the temporary loss of consciousness [6]. A CLR was found in 13–30% of subjects with NDEs, leaving an impression of countless images of everything the person had experienced from early childhood to the present [7,8].

The data on CLR phenomenology are extremely mixed. For instance, Katz et al. qualitatively analyzed in-depth interviews of seven volunteers who had self-reported a CLR [9]. The authors demonstrated that CLR experiencers were equally likely to report their CLR in chronological, associative, or random order. Similarly, they described the CLR as successive or simultaneous and indicated either a first- or third-person perspective with the same frequency. Neither an affective tone nor recurrent themes in CLRs were found to be consistent.

The notion of life “flashing” before our eyes in the face of any danger has been widely assimilated by popular culture. This idea is featured in numerous works of literature, film, cartoons, and anecdotes, such as Sam Mendes’s “American Beauty” (1999), Cameron Crowe’s “Vanilla Sky” (2001), and Danny Boyle’s “127 Hours” (2010).

What does the phenomenon of the CLR really stand for? Is it just a metaphor to express personal experience beyond description? Alternatively, is it a real experience of sped-up AM functioning? The CLR may be also seen as an example of false memories, similar to dream memories or memories of stressful events [10,11].

The scientific research on CLRs in real-life situations has been confounded both by methodological limitations and trivial ethical reasons. To date, the methodology of CLR research is limited to retrospective self-reports. As a result, we know almost nothing about the cognitive processes that underlie the CLR, its neural basis, or its evolutionary importance. Due to this reason, this study was driven to introduce an alternative methodology. Namely, it was aimed to develop a procedure for eliciting the CLR-like experience in a laboratory setting and then examining different aspects of the “here and now” perspective. Importantly, this technique should be effective and safe for participants.

### Justification of Technique for Eliciting CLR Experience

The main cognitive paradox of the CLR experience is that the total recall of the personal past should be provided by working memory (WM), which is a term often used almost interchangeably with consciousness. Although this equation may be questioned [12], WM is assumed to be extremely capacity-limited in regard to both the number of information units and the processing speed [13]. Due to the limited capacity of WM, it seems technically impossible to be aware of numerous memories simultaneously. Consequently, sometimes authors have argued that the CLR provides evidence of the mind being separate from the body and brain, or even the persistence of life after death [14]. From my perspective, these parapsychological interpretations originate from the shortage of self-report methodology and the ignorance of the recent developments in memory models; therefore, a new approach to the theory of CLR content and mechanisms is required.

In the search for the theoretical background for the laboratory paradigm modeling of the CLR experience, it seems essential to find arguments that WM is potentially able to support this kind of experience and to propose a novel view of the CLR experience.

In the WM research field, many influential theorists have considered WM limitations as a surmountable property, at least to some extent. First, instead of a limitation in the absolute amount of information that can be maintained in WM at once, people in an ordinary state of mind are able to simultaneously process about six acoustically encoded units of information [15] and about four visually encoded units of information [16]. Although having a fixed number of slots is an obvious limitation, the amount of information held by WM could be dramatically increased by combining numerous elements of information into one integrated unit. Second, whereas strict WM limitations are crucial for the processing of novel information for the first time, this does not apply absolutely to the retrieval of well-learned material held in the long-term memory (LTM) [17]. Due to the network structure of LTM and the schematic form of event representation in LTM, “high element interactivity material that has been incorporated into an automated schema after extensive learning can be manipulated easily in WM” [17], p. 222. Third, solving complex problems is possible without a complete awareness of all relevant information. For instance, in Ericsson and Kintsch’s long-term WM model [18], the strong association between the schema and a sufficient retrieval cue may serve to extend the effective WM capacity in people who have domain knowledge. According to this model, once a strong association between the cue and the schema is established, this cue becomes its “legal representative” in WM, since there is no longer any necessity to retain the whole unit of information in WM. The authors formulated the point as, “At the time of recall, only the node corresponding to this specific structure (schema) needs to be available in WM, along with the retrieval cue specifying the type of desired information” [18], p. 10. More radically, in concentric or activation-based models of WM proposed by Cowan [19] and Oberauer [20], WM and LTM differ from each other in their states of representation, not structurally. These differences may be described in four regions: focus of attention (FA), direct-access region (DAR), activated part of LTM (aLTM) and the rest of LTM in the passive state. These models indicate that the processing of activated items outside the DAR of WM is possible without complete awareness via associated links. Therefore, similar to the LT-WM model, the state of consciousness may be affected by a set of activated LTM items that are not present in WM.

According to the hypothesis, the CLR experience is a spontaneous AM illusion of simultaneous access to one’s entire personal past, not real access to its chronological day-by-day mental representation. This illusion evokes a retrospective self-reported experience of the total recall of one’s personal past. Objectively, this may be helpful for the mobilization of an experiencer by enhancing their feelings of self-continuity and, hence, leading to an increase in the probability of coping with a life challenging/threatening situation.

Since the boosting of subjective self-continuity was addressed as the main function of the CLR experience, not all autobiographical memories are suitable as functional components of the CLR illusion. It seems reasonable to focus on a set of highly significant, personal, self-defining memories (SDM), those that contribute most heavily to humans’ overall sense of self [21,22]. SDMs are products of elaborative rehearsal, not just of occasional encoding. SDMs combine rich perceptual details of the experienced events with a process of meaning-making. In daily activities, there is no need to reconfirm one’s identity, values, and goals, so SDMs might be inconspicuous. When life-threatening events disrupt the everyday routine, a retrieval mode for SDMs is activated, triggering the pursuit of rescue [23]. Notably, SDMs are linked to thematically similar memories that share the same theme. This means that SDMs may be considered a top node of the hierarchical network. These properties (semantic elaboration of experience, relevance to a feeling of self-continuity, automated retrieval in response to the goal–activity frustration, and the top position in the memory hierarchy) make SDMs the most suitable candidates for components of the CLR illusion. Moreover, it may be assumed that well-trained SDMs appropriately meet all the criteria for enhancing WM capacity.

Following from the WM models mentioned above, the CLR illusion does not require the immediate replay in the WM of numerous representations of the past, which were gradually accumulated in LTM for decades. I speculate that, when a CLR occurs in a real-life setting, it may consist of almost parallel partial access to a set of SDMs, without a complete awareness of any concrete episodes. In this process, each SDM occupies one slot of WM space and serves as a node in the entire network of relevant autobiographical experiences. This results in illusion of massive, seemingly unlimited, amounts of autobiographical information that are later collectively labeled as a CLR. In the face of real danger, the survival motif is an appropriate cue for the manifestation of SDM retrieval. This is not the case in a safe laboratory environment. As a result, in the study reported below, the participants were provided with artificial cues to engage SDMs with the DAR of WM. Consequently, it was supposed that the parallel awareness of a small set of verbal associative cues to corresponding SDMs would be sufficient to provide a target phenomenon for the CLR experience.

The main objective of this work was to examine a novel method for eliciting the CLR experience introduced above in healthy participants in two complimentary aspects. Firstly, it focused on both self-reported evidence of the target CLR-like experience and objective data on eye movements accompanying the reported experience. For this purpose, a complex of typical oculomotor reactions accompanying SDMs when the latter appear in response to well-trained verbal cues has been identified. This complex was compared with previously obtained data on a similar topic [24,25]. The main part of the experiment was administered with the aim of matching self-reports about experiencing the simultaneous audio presentation of SDM cues (SAPSDMC) technique and eye-tracking data. The hypothesis was that the SAPSDMC technique would evoke the self-reported CLR-like experience and produce a pattern of eye movements similar to those identified in a well-trained, cued SDM retrieval.

## 2. Materials and Methods

### 2.1. Participants

Twenty graduate students or employees at the Lomonosov Moscow State University (Moscow, Russia) participated in the study (Mean age = 26.95 years, SD = 4.64, range 23–35 years; 8 men). They were native Russian speakers with normal or corrected-to-normal vision and hearing; all were right-handed. All subjects provided their informed consent for inclusion before participating in the study. The study was conducted in accordance with the Declaration of Helsinki, and the protocol was approved by the Ethics Committee of the Faculty of Psychology of Lomonosov Moscow State University (Project No. 2018/27).

### 2.2. Apparatus

For the eye-tracking experiment, an iView X™ HI-SPEED SMI^®^, was used as a desktop system monocular eye tracker. This system is based on the dark pupil system, through which the eye is illuminated by an infrared light. Once the calibration was satisfactorily completed, an accurate estimation of the gaze location could be acquired. The sampling rate of this system is 50 Hz and its gaze position accuracy is 0.5°–1.0° (typ.) according to the manufacturer. During the experiment, the participants were free to explore all parts of the blank-displayed screen, and asked not to look outside the screen, and their gaze location was recorded by the eye tracker. The distance between participants and the screen was 70 cm and, behind the screen, there was a blank white wall displaying no visual stimuli.

### 2.3. Materials

The CLR imitation technique, termed SAPSDMC, includes five stages:Each participant recollected 8 SDMs, prioritizing memories addressing different life themes and ages. The exact instruction was as follows: “Recollect, in as much detail as possible, eight episodes from your past that represent to the maximum degree your personal traits and illustrate what kind of person you are”;Each SDM was marked with a unique two-word title cue;The association between the target SDM and the cue title was tested three times successively. In each trial, the participants were told to feel free to allocate as much time to complete the visual image as they needed;Each SDM cue title was audio recorded separately, while a participant loudly repeated it for 30 s;The audio recordings were superimposed on each other with a digital music editor, AdobePremiereProCS3 Portable.exe. As a result, all SDM cue titles could be perceived by the listener simultaneously. It sounded like a choir of verbal cues of personal memories.

For the control conditions, each participant loudly read a list of neutral phrases not associated with autobiographical memories. Then, they were elaborated as described above.

### 2.4. Procedure

The participants participated in two sessions over 8–10 days. In the first session, materials for SAPSDMC were collected. In the second session, the eye movements of 20 participants were recorded in four conditions. The recording lasted 30 s for each condition, with the immediate collection of self-reports following 1 min rest before the next condition started. All conditions followed one after another in the order of Conditions 1, 2, 3, and 4. The scheme of the experimental procedure is presented in Figure 1.

In Condition 1, the participants were eye-tracked while listening to the 30 s repetition of one isolated stimulus consisting of the two-word neutral phrase. In Condition 2, the participants were eye-tracked while listening to the 30 s repetition of one isolated stimulus consisting of the well-trained cue title that referred to a randomly selected SDM. In Condition 3, the participants were eye-tracked while listening to the 30 s repetition of eight superimposed two-word neutral phrases. In Condition 4, the participants were eye-tracked while listening to the 30 s repetition of eight superimposed autobiographical audio tracks referring to SDMs. In all conditions, the participants had to look at a blank screen while their gaze location was recorded by the eye-tracker. Immediately after each presentation, self-reports were collected. The participant received the following instruction “Describe everything you experienced during the presentation. You are free to talk about your thoughts, memories, emotions, or body sensations, etc.” None of the participants were informed that the study was relevant to the CLR phenomenon.

## 3. Results

### 3.1. Self-Reports

All participants reported that in response to the single cue title, the appropriate SDM came to mind in the form of a visual image. Listening to the single neutral two-word phrase predominantly left an impression of converting a speech into a mental picture. However, seven participants indicated that they did not experience any mental imagery, and were simply listening to the speech.

None of the participants reported a CLR-like experience after listening to the repetition of eight superimposed two-word neutral phrases. In contrast, the most common description of the experience was “noise”. Listening to the repetition of eight superimposed autobiographical audio tracks provoked different reactions. Half of participants did not experience any component of a CLR, perceiving the stimulus similarly to the relevant control condition. The remaining 10 participants reported a CLR-like mnemonic illusion that included: unusual state of mind, underestimation of time passage, physiological arousal, impression of “all memories at once”, vivid memories not recalled in the preparatory session, and positive emotions. Seven participants used metaphors, such as “a funnel”, “a tunnel” and “a safe capsule” to describe their experiences. Four subjects declared that they had a СLR experience in their past, noting the similarity of the naturally occurring and laboratory experiences.

The self-reports, after listening to the eight superimposed stimuli, are summarized in Table 1.

### 3.2. Eye Tracking Results

The following five eye movement variables were included in the analysis: mean frequency of fixations per minute, mean fixation duration (ms), mean frequency of saccades per minute, mean saccade duration (ms) and scanpath length (px). The conditions consisting of a single stimulus presentation (first and second) and the conditions consisting of eight superimposed stimuli (third and fourth) were compared in pairs.

Listening to a single audio reference to an SDM, in comparison with a two-word neutral phrase, triggered a lower frequency of fixations (*F* (1,19) = 19.958, *p* < 0.000, MSE = 0.07, η² = 0.512), a longer fixation duration (*F* (1,19) = 20.243, *p* < 0.000, MSE = 152737, η² = 0.516), a lower frequency of saccades (*F* (1,19) = 10.307, *p* = 0.005, MSE = 0.09, η² = 0.352), and a shorter scanpath length (*F* (1,19) = 6.419, *p* = 0.02, MSE = 2,207,899, η² = 0.253). Taken together, these data indicate that SDM retrieval may be associated with oculomotor immobilization.

In the next step of the analysis, I contrasted the eye-tracking data recorded while listening to the eight superimposed stimuli between two groups: CLR experiencers (*n* = 10) and non-experiencers (*n* = 10). The groups were divided on the basis of the self-reports. A liberal criterion for assigning participants into the “CLR experiencers” group was applied. Due to the exploratory nature of the study, I considered the presence of at least one of the components of the CLR experience in the self-report sufficient for labeling the experience as CLR-like.

A 2 × 2 (condition: autobiographical, neutral × group: CLR experiencers, non-experiencers) ANOVA revealed significant condition × group interactions for the frequency of fixations (*F* (1,18) = 6.372, *p* = 0.021, MSE = 0.283, η² = 0.261), fixation duration (*F* (1,18) = 4.405, *p* = 0.046, MSE = 1,977,926, η² = 0.203), frequency of saccades (F (1,18) = 5.038, *p* = 0.038, MSE = 0.236, η² = 0.219) and scanpath length (*F* (1,18) = 4.405, *p* = 0.05, MSE = 1.100E7, η² = 0.197).

In CLR experiencers, the presentation of eight superimposed autobiographical audio tracks generated fewer fixations (*F* (1,9) = 15.312, *p* = 0.004, MSE = 576,000, η² = 0.630), a longer fixation duration (*F* (1,9) = 5.710, *p* = 0.041, MSE = 2,229,977,545, η² = 0.388), fewer saccades (*F* (1,9) = 13.440, *p* = 0.005, MSE = 376,200, η² = 0.599), a shorter saccade duration (*F* (1,9) = 7.040, *p* = 0.026, MSE = 178,886, η² = 0.439) and a shorter scanpath length (*F* (1,9) = 12.127, *p* = 0.007, MSE = 6,204,906,689, η² = 0.574) in comparison with the control “noise” condition. A self-reported CLR-like experience, as assessed by eye-tracking, was associated with visual immobilization.

In contrast, no differences were observed in the fixation count (*F* (1,9) = 0.276, *p* = 0.612, MSE = 1,465,000, η² = 0.03), fixation duration (*F* (1,9) = 0.276, *p* = 0.612, MSE = 1,725,875,566, η² = 0.03), saccade count (*F* (1,9) = 0.348, *p* = 0.570, MSE = 1,324,800, η² = 0.037), saccade duration (*F* (1,9) = 0.495, *p* = 0.500, MSE = 57,738, η² = 0.052) and scanpath length (*F* (1,9) = 0.086, *p* = 0.775, MSE = 1.579 × 10 ⁷, η² = 0.010) between experimental and “noise” conditions in participants who reported no CLR-like experience.

The results of the eye-tracking experiment are summarized in Table 2.

## 4. Discussion

The CLR experience may be defined as a mnemonic illusion of total recall. Although the data on CLRs are exclusively regarded as retrospective, with self-reports of NDEs generally being provided by those who have suffered from cardiac arrest or head injury survivors [24,25], CLR-like accounts were found to be normally distributed in the general population when assessed by quantitative psychometric scales [9].

This current study was guided by the theoretical assumption that, contrary to naïve popular belief, a subjective CLR experience does not require an instant retrieval of countless memories. A concentric or activation-based model of memory was taken as a promising theoretical framework for an alternative model of CLR genesis. Based on these models, I propose that a sufficient condition for experiencing a CLR is a short-term and insight-like access to the activated LTM traces, without loading them to the WM region. In other words, the subjective rapid flow of images depicting a subject’s entire life may arise as a feeling of potential accessibility to past episodes that are essential for self-coherence and self-identity. This type of memory is known as SDM. Notably, due to the automatic and unintentional nature of the CLR phenomenon, it does not require the effortful process of visual exploration that is typically detected during autobiographical retrieval. As CLRs naturally occur in life-threatening situations requiring many resources for survival, only those SDMs that are already well established and ready to be retrieved may serve as candidates for CLR content.

Like other extreme human experiences, the CLR is difficult to study under controlled laboratory conditions, given the ethical concerns and more stringent requirements applied to scientific procedures (e.g., the requirement of reproducibility and replicability). Due to this reason, I attempted to develop a safe and well-controlled procedure for evoking a CLR-like experience. In order to do so, I avoided using psychoactive compounds—such as those used in the ketamine model of the NDE [26]—and inflicting any prospective harm to participants. The resulting technique is termed simultaneous audio presentation of self-defining memory cues (SAPSDMC).

Since nothing is known about specific eye movement patterns during the recall of well-prepared SDMs, the first stage in this study was to determine whether the automatic recall of a single self-defining autobiographical episode, previously trained in response to an audio cue evokes a distinguishable eye movement signature.

Previously reported results indicated a lower number of fixations but a larger number, duration, and amplitude of saccades during autobiographical retrieval in comparison with counting [27], and a larger number of fixations and saccades, but a shorter fixation duration, during the retrieval of highly emotional autobiographical memories in comparison with neutral memories [28]. In contrast, the results showed a lower fixation count together with a longer fixation duration, a lower saccade count, and a shorter scanpath length in SDM compared to the neutral control phrase.

Although all participants reported that the target memory came to their minds in the form of a visual image, their eyes remained relatively calm during the autobiographical stimulus presentation. Simultaneously, the eye-tracking detected high eye motility in response to the neutral phrase, which, according to self-reports, caused a relevant mental picture. These findings reveal that the awareness of the previously trained self-defining autobiographical episode did not evoke the active visual exploration that is typically associated with retrieval of an AM. Hence, I assume that, at least in the case where a mental image has been carefully constructed in advance, the necessity of visual activity may be called into question. These findings might be also relevant to the speculation surrounding the mandatory preparedness of CLR memories discussed above.

In the second stage of this study, I examined the subjective state of mind in participants while listening to eight superimposed audio tracks consisting of verbal cues to SDMs. I hypothesized that this procedure would induce an experience similar to a naturally occurring CLR. Half of participants reported some of the core components of the CLR experience after undergoing the SAPSDMC procedure. In these participants, the SAPSDMC procedure and SDM retrieval in response to a previously trained verbal cue triggered a similar pattern of eye movements. Namely, in CLR experiencers, the presentation of eight superimposed SDM audio tracks generated fewer fixations, a longer fixation duration, fewer saccades, a shorter saccade duration and a shorter scanpath length in comparison with the control noise condition that consisted of eight superimposed two-word neutral phrases. A significant similarity in eye movements was found between responses to a single cue to an SDM and a choir of verbal references to a set of SDMs; I attributed the results to the proximity of the mechanisms behind both experiences.

## 5. Conclusions

At the outset of the present study, I recognized that the CLR experience could not be explored satisfactorily on the basis of the common, exclusively retrospective self-reporting methodology. Hence, in this field of research, I recognized the need for an alternative methodology that could elicit the target phenomenon in the laboratory with the aim of investigating CLRs “here and now”.

The main contradiction of the CLR experience is that total recall should be provided by the capacity-limited working memory (WM). Notwithstanding, from the perspective of long-term working memory, concentric and activation-based models of memory, this obstacle seems surmountable. Following the results of these models, a technique of superimposing audio recordings of previously trained verbal cues to SDMs, i.e., memories essential for self-identity and self-coherence, was developed. This technique proved to be effective in evoking the core components of a CLR-like mnemonic illusion in 50% of participants. The objective of the empirical study was to obtain eye-tracking data that supported the hypothesis that eye movements during self-reported components of the artificial CLR-like experience in response to a choir of SDMs were similar to eye movements during the retrieval of a single previously trained SDM. This was found to be the case in CLR experiencers. I consider this similarity as evidence for the statement that partial access to SDMs produces the target CLR-like phenomenon.

Due to the innovative nature of the reported study, some of its limitations should be underscored. First, the sample included volunteers who had not been interviewed about their previous CLR experiences. Incidentally, four subjects declared that they had a СLR experience in the past, noting the similarity of the naturally occurring and laboratory experiences. Future research should be conducted using a sample of CLR experiencers to explicitly compare both states of mind. Second, our understanding of SDMs must be advanced. Instead of the direct instruction typically employed in SDM studies, the SDMs could be selected more carefully via individually relevant cues. Third, limiting the number of SDMs in the SAPSDMC technique to six, with a control for equitable temporal distribution, appears promising. Finally, although I have speculated on the potential adaptive function of the CLR, further research is required to fully establish the validity of this proposed function. Future work could examine the impact of a CLR-like laboratory experience on performance in complex and challenging tasks, possibly in a virtual reality environment. From my perspective, the findings open empirical avenues for examining the potential causal relationship between a CLR-like experience and significant increases in cognitive performance.

## Figures and Tables

**Figure 1 behavsci-10-00060-f001:**
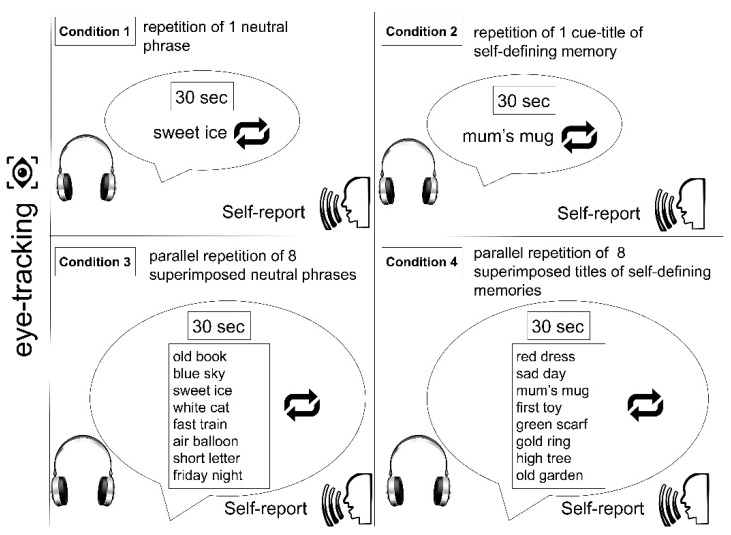
A schematic illustration of the experimental protocol. The conditions were separated by self-reports and one minute of rest.

**Table 1 behavsci-10-00060-t001:** Self-reports (excerpts) after listening to eight superimposed self-defining memories (SDM) cues or to eight superimposed neutral phrases (direct quotes translated from Russian). The participants in the “compressed life review (CLR) experiencers” group are marked by an asterisk.

No.	8 Superimposed SDM Cues	8 Superimposed Neutral Cues
1 *	Severe dizziness, like on the carousel and then trembling. After a while, a strong desire for activity.	Only noise
2	Changes in rhythm were perceived as music, which was not disturbing, but like a radio far away. Did not understand what the program was about, but thought the voices were soothing.	Imperceptible bunch of sounds.
3	Similar to the previous one. Maybe this time I was prepared, so it was not that frustrating.	Almost vomiting, very unpleasant feeling.
4	Completely new set of words. I guess that there were a couple of my memories’ titles. Am I right?	I heard few words, namely “ice”, “morning”. I forgot the rest.
5 *	The sensation appeared as in the childhood, when you leave your body and begin to unwind, and then spin in the other direction. Like in a funnel. From the top to the bottom. Something felt bursting from the inside.	Noise, it felt like being about a catastrophic headache.
6	Made me sleepy. Might be tired from the long session. My voice, as well. My voice transformed by some specific software.	I recognized my own voice, but the words sounded lengthy.
7 *	Somewhere about the middle of the track I was struck by the pleasant thought that all this was with me, and that all this was me and about me. Pictures began to appear. I think these were memories, but I cannot describe specific scenes. If I need to give a metaphor…a tunnel, a huge luminous tunnel, and I was a small crumb inside it.	I tried to differentiate words and phrases. Not very successfully. I expected that you would ask me later about how many phrases were presented.
8 *	I did not perceive any memories from this group of words. Maybe once, but I’m not sure it was because of the track. Suddenly, I had a thought about my grandma. I really, really miss her. Wait! I had more. A chain of memories. The scene when a saw my grandma for the last time. I now can recall many.	I was surprised by the strange sounds. It was like a noise in the forest, perhaps the cries of wild animals. For a moment, I was scared.
9	I was pretty sure that these were my phrases on top of one another, which was much more clear than earlier. My voices seemed to demand something from me. They were angry. For a moment, I felt guilty.	It reminded me of an orchestra setting before a concert. I assumed I would hear my voice, but it did not sound like my voice.
10	I immediately heard the phrase “black chestnuts” and then I heard only that as the main voice. Other voices seemed quieter and sounded background. It seemed a bit longer than the previous “choir”, but definitely much longer than single repeating phrases. Maybe too loud.	In a couple of seconds, I lost the sense of what was going on.
11	A kind of mantra. Could this be my personal mantra? I need to try a couple more times.	Shamanic tune. So they probably call for rain.
12 *	No time. No time at all. I dived deeply and stayed there. A peace, calm, eternity. My past and my future both inside and outside me.	Strange feeling of empty mind. Tried to think about something and found out that it was impossible. So just waited for the end.
13	I can say the same I already said about the previous audio. Does not make sense to me.	A cow mooing sound. A herd of cows. Seems it was long. Endured to the end just because you asked me.
14 *	I felt like I jumped into a tunnel through the center of a huge mountain. The scenes circled around me extremely fast, but I could snatch out single words.	Bad choir without harmony. Maybe I needed more time to get a taste for it.
15 *	When I became able to adjust my hearing and joined the situation, I noticed that I stopped listening and started remembering that day. I stopped hearing that something was playing in the headsets. I felt again that I was a little girl who was going to her friend to stay with her so that she would not be bored and scared of being alone at home.	As if chirping birds. Loud twitter in spring street. I did not immediately understand that these were human voices. A very strange feeling.
16 *	What a pleasure this new feeling! Feeling of internal integrity, completeness. It started when I understood that it was my life, the precious fragments of my life in once, with me now. I noticed that I was smiling while remembering this. I felt completely inside my memories, like in a safe capsule.	The chatter of unfamiliar voices. Sometimes separate words and exclamations erupted. I did not know how to figure it out.
17	Not so disturbing as the previous one. Maybe it was shorter. No special sensation, except I am bit tired.	Nothing to say. Just a disturbing noise.
18 *	Cosmic feeling. I felt like an astronaut who is connected with a spaceship and at the same time flies into open space. It was as if I were looking at the Earth from space. Only instead of the Earth I saw my life.	These were notes of my words superimposed on each other in parallel. Choir of my voices. I felt surprised.
19	The same mumble, but only more persistent. I was no longer able to be distracted. Therefore, I had to listen to the end.	It was like senile rushing. At first I listened, but then I tried to distract myself, because all was the same, nothing was clear.
20 *	Pictures began to appear in my mind of how we played with my friends, of how we imagined ourselves as “adult aunts” who were about to relax over a cup of tea, of how we dreamed and imagined our future. These pictures did not stay in one place, they moved into a kind of funnel.	Passages of phrases succeeded each other extremely quickly. One phrase “midnight conversation” seemed to me pronounced in a different voice and I began to repeat it until the end of listening.

**Table 2 behavsci-10-00060-t002:** Eye movement variations during listening to neutral phrases (single or eight superimposed) or listening to SDM cues (single or eight superimposed).

	Single SDM Cue	Single Neutral Cue	*p*	Superimposed SDM Cues		*p*	Superimposed Neutral Cues		*p*
				CLR Experiencers	Non-Experiencers		CLR Experiencers	Non-Experiencers	
Fixation frequency per min	48.000 (31.868)	70.500 (38.140)	**	44.400 (40.418)	84.000 (44.899)	*	86.400 (49.574)	75.000 (49.173)	ns
Fixation duration (ms)	1864.180 (1619.965)	1308.135 (1586.248)	**	2643.820 (2280.761)	1019.280 (1014.416)	*	1048.000 (1059.504)	1328.100 (1430.988)	ns
Saccade frequency per min	29.700 (25.933)	48.000 (33.717)	*	33.000 (34.438)	67.800 (39.703)	*	64.800 (43.062)	58.200 (42.619)	ns
Duration of saccades (ms)	32.810 (14.090)	36.325 (12.286)	ns	26.030 (18.716)	37.850 (14.985)	*	41.900 (11.722)	40.240 (7.598)	ns
Scanpath length in px.	1839.000 (3518.358)	3029.450 (3281.443)	*	1446.900 (1789.653)	3769.400 (3462.493)	**	5326.300 (4867.091)	3247.000 (5215.510) ^1^	ns

^1^ Standard deviations are provided in parentheses. The difference within the following corresponding conditions were significant for intragroup comparison at * *p* < 0.05 and ** < 0.01.

## Data Availability

The datasets used and analyzed during the current study are available from the corresponding author on reasonable request.

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
