# Peer review of "Compressed Life Review: Extreme Manifestation of Autobiographical Memory in Eye-Tracker"

_behavsci, 2020, doi:10.3390/bs10030060_

Round 1

Reviewer 1 Report

This is a very original study presenting a new technique for objectively studying the compressed life review effect, whereby ones whole life "flashes before ones eyes". Some interesting findings are presented using eye movement measures which compare eye movement activity during viewing of a blank screen following cues designed to elicit a CLR experience, and neutral cues without any memory associations. fMRI scanning was also carried out although methods and results are not reported in detail for the neuroimaging data.

I have some methodological criticisms / areas where clarification is needed:

It would be helpful to include a schematic illustrating the task sequence / protocol. Giving examples of the types of words associated with memories or not and the timings of the stimulus presentation and eye movement recording periods under the different conditions. At the end of the methods it suggests that only participants that experienced the CLR effect were selected. It is crucial to provide more information on how participants were selected for further analysis and how it was determined whether or not they had experienced the CLR effect. What criteria were used to make this decision? The description of fMRI methods, analysis and results is unsatisfactory. What was the total duration of fMRI acquisition? The task seems very short for fMRI analysis as the CLR effect is (I assume) something that happens within seconds. What software / statistics were used to analyse the fMRI data? Provide results in the form of an activation table with p values for any contrasts run and possibly figures showing activation differences between conditions.

Alternatively if more detail / justification is not available for fMRI the authors may consider removing the fMRI component from the paper and just concentrating on eye movements.

Author Response

Sincerely,

prof.Veronika Nourkova

Reviewer 2 Report

The current study aimed to develop the first lab-based experimental technique for eliciting the Compressed Life Review (CLR) in no risk participants. The technique was able to evoke CLR-like experience in one half of the 20 participants. Similar eye-movement patterns were observed between a single SDM condition and a “choir” of SDMs condition in self-reported CLR experiences.

The current manuscript suffers from several serious issues. Overall, I found the manuscript not very well structured. Some critical information are missing (see my comments below) some sections are misplaced (e.g. discussion in the results section), preventing the reader from understanding properly of the manuscript.

Reading the introduction, I had the impression that the author makes a lot of speculative assumptions to develop their lab paradigm modelling CLR (eg. line 81). More references to literature would strengthen the author hypothesis and the validity of her experimental technique.

The main aim of the present study is a bit vague, “examine the novel method for eliciting CLR”. For instance, was the eye-tracking method used as a validation tool to show that the “choir” of SDMs triggers the same pattern of visual exploration than a single SDM? The different goals

One major issue, according to me, is the absence of method and results sections regarding the self-reported data. Although it is stated at the end of the introduction that the main aim of the study is “to explore the novel method for eliciting CLE experiences with self-reports and eye-tracking, fMRI technics”, there is absolutely no method description of the self-reported measures (questionnaires, scales…), nor results section or statistical analysis.

Relatedly, in the results section, participants’ self-reports should be reported in a separate subsection and not in the eye-tracking experiment subsection.

Materials and Methods section:

The description of “1-1” and “8-8” conditions is unclear and difficult to understand at first read.

The author wrote “we predicted that only part of the sample would report CLR experience in 8-8 condition”, on what basis this assumption was made?

Results section:

How the two groups (CLR experiencer and CLR non-experiencer) were divided, based on which questions? Which score? Do these two groups differ with regard to the self-reports above-mentioned? Statistics should be provided for the phenomenological data.

Table 1: the author should report the statistics correctly, the star corresponding to the p-value should not be associated to the mean of one group, it doesn’t make sense.

3.2 fMRI experiment

Line 207: “The group analysis…” Which group analysis is the author talking about? The reader doesn’t have enough information for proper understanding.

The second (line 213) and third paragraphs (line 218) should not be included in the results section, as they discuss the fMRI results with regard to literature, they should be part of the discussion instead. In addition, there is currently no discussion/interpretation of the fMRI results in the discussion section.

The fMRI section needs much more elaboration on the method used and the analysis conducted. It is a serious weakness on the present manuscript.

Minor comments

Line 76: “According to our hypothesis” I wonder who is “our” since there is only one author.

Line 77: It might be a mistake for reference [12]

Round 2

Reviewer 1 Report

Although some changes have been made to improve this manuscript I feel there are some fundamental issues which will be difficult to resolve.

The connection between the eye movement results / analysis and the bigger theoretical picture (activation-based model of memory) being presented is not clearly explained. The theoretical accounts and relationship between CLE and SDM outlined at the start of the discussion lack clarity and grounding in the presented data. The issue of how participant experiences were classified as being CLE phenomena or not has not been adequately addressed and there do not seem to have been very objective criteria used for grouping participants in the two groups. There is therefore a danger of experimenter bias influencing resulting data analysis due to this subjective process.

A lot of use of personal pronouns "I" / "we" occurs throughout the manuscript and emphasises the personal theoretical perspective being expounded.

Reviewer 2 Report

The manuscript has substantially improved and is much clearer now.

The author has answered all my inquiries.

Before recommending the present article for publication I strongly encourage the author to perform an extensive English proof-reading for language and style.

I also suggest to add a perspective section in the discussion to open new research questions based on the new lab paradigm developed by the author.
